# PET Probes for Preclinical Imaging of GRPR-Positive Prostate Cancer: Comparative Preclinical Study of [^68^Ga]Ga-NODAGA-AMBA and [^44^Sc]Sc-NODAGA-AMBA

**DOI:** 10.3390/ijms231710061

**Published:** 2022-09-02

**Authors:** Ibolya Kálmán-Szabó, Judit P. Szabó, Viktória Arató, Noémi Dénes, Gábor Opposits, István Jószai, István Kertész, Zita Képes, Anikó Fekete, Dezső Szikra, István Hajdu, György Trencsényi

**Affiliations:** 1Division of Nuclear Medicine and Translational Imaging, Department of Medical Imaging, Faculty of Medicine, University of Debrecen, Nagyerdei St. 98, H-4032 Debrecen, Hungary; 2Gyula Petrányi Doctoral School of Clinical Immunology and Allergology, Faculty of Medicine, University of Debrecen, Nagyerdei St. 98, H-4032 Debrecen, Hungary; 3Doctoral School of Clinical Medicine, Faculty of Medicine, University of Debrecen, Nagyerdei St. 98, H-4032 Debrecen, Hungary; 4Doctoral School of Pharmaceutical Sciences, University of Debrecen, Nagyerdei St. 98, H-4032 Debrecen, Hungary

**Keywords:** [^44^Sc]Sc-NODAGA-AMBA, [^68^Ga]Ga-NODAGA-AMBA, gastrin-releasing peptide receptor (GRPR), bombesin (BBN), prostate cancer (PCa), PC-3, positron emission tomography (PET)

## Abstract

Gastrin-releasing peptide receptors (GRPR) are overexpressed in prostate cancer (PCa). Since bombesin analogue aminobenzoic-acid (AMBA) binds to GRPR with high affinity, scandium-44 conjugated AMBA is a promising radiotracer in the PET diagnostics of GRPR positive tumors. Herein, the GRPR specificity of the newly synthetized [^44^Sc]Sc-NODAGA-AMBA was investigated in vitro and in vivo applying PCa PC-3 xenograft. After the in-vitro assessment of receptor binding, PC-3 tumor-bearing mice were injected with [^44^Sc]Sc/[^68^Ga]Ga-NODAGA-AMBA (in blocking studies with bombesin) and in-vivo PET examinations were performed to determine the radiotracer uptake in standardized uptake values (SUV). ^44^Sc/^68^Ga-labelled NODAGA-AMBA was produced with high molar activity (approx. 20 GBq/µmoL) and excellent radiochemical purity. The in-vitro accumulation of [^44^Sc]Sc-NODAGA-AMBA in PC-3 cells was approximately 25-fold higher than that of the control HaCaT cells. Relatively higher uptake was found in vitro, ex vivo, and in vivo in the same tumor with the ^44^Sc-labelled probe compared to [^68^Ga]Ga-NODAGA-AMBA. The GRPR specificity of [^44^Sc]Sc-NODAGA-AMBA was confirmed by significantly (*p* ≤ 0.01) decreased %ID and SUV values in PC-3 tumors after bombesin pretreatment. The outstanding binding properties of the novel [^44^Sc]Sc-NODAGA-AMBA to GRPR outlines its potential to be a valuable radiotracer in the imaging of GRPR-positive PCa.

## 1. Introduction

Given the immense burden entailed by the rising prevalence of prostate cancer (PCa), the necessity of the introduction of such imaging modalities that excel in timely diagnostic assessment of primary tumors, as well as recurrent diseases, is highlighted. Novel PET imaging modalities seem to be promising in the evaluation of localized primary PCa, follow-up, and the re-emergence of the neoplasm [1,2].

Since gastrin-releasing peptide receptors (GRPR)—overexpressed in malignant epithelial prostate cells—are assumed to be valuable biomarkers, GRPR-targeted molecular PET imaging may widen the diagnostic armamentarium of PCa [3,4,5,6]. Fourteen amino-acid based bombesin (BBN) is a gastrin-releasing peptide (GRP) analogue exerting high affinity and specificity to GRPR [7]. Accumulating research studies report about radiolabeled GRPR-ligands, including bombesin (BBN) and its analogues in GRPR-associated PCa imaging [8,9]. Prior literature data state that aminobenzoic-acid (Gly-4-Abz-Gln-Trp-Ala-Val-Gly-His-Leu-Met-NH_2_; AMBA) could be a potentially successful synthetic BBN analogue in the isotope diagnostics of PCa [10].

Several radiometals are available for the labelling of peptide-based radiopharmaceuticals. The following appropriate physical features of ^18^F led to its widespread clinical application in cancer diagnosis: 100% positron efficiency, 0.64 MeV energy, and t_½_ 109.7 min [2]. Besides ^18^F, ^64^Cu also represents an area of growing investigation in terms of peptide-labelling, however, its lower obtainability, prolonged half-life, and enhanced radiation danger are those shortcomings that need to be addressed [2,11]. Despite its disadvantages, [^64^Cu]Cu-labelled GRPR antagonist D-Phe-Gln-Trp-Ala-VaI-Gly-His-Sta-Leu-NH_2_ conjugated either to NOTA or NODAGA was reported to possess considerable value in the PET imaging of GRPR-expressing malignancies at preclinical level [12]. ^18^F-labelled BBN analogues AMBA and RMI (DOTA-CH_2_CO-G-4-aminobenzoyl-f-W-A-V-G-H-Sta-L-NH_2_, GRPR antagonist) were first synthesized for molecular PET imaging of PCa in 2013 [2]. Positron emitter 68-Gallium (^68^Ga) eluated from readily available 68-Germanium/68-Gallium (68Ge/68Ga) generators has favorable energy, half-life, chemical purity as well as quality that are satisfactory for the radiolabeling process of different peptides including AMBA [7,13,14]. Suitable targeting properties of ^68^Ga- or 177-Lutetium (^177^Lu)-labelled, DOTA-conjugated AMBA were confirmed in preclinical studies [10,15]. Further, ^68^Ga-AMBA PET was found to be better than ^18^F-methylcolin-based metabolic in-vivo PET diagnostics of PCa [16]. GRPR targeting potential of ^68^Ga- and ^177^Lu-labelled ProBOMB2—a novel BBN derivate—was investigated in preclinical models of GRPR-positive PC-3 human prostate cancer tumor-bearing male immunocompromised mice [17]. Based on the high-quality images of exceptional contrast and discrete background activity, as well as satisfactory pharmacokinetic properties, these novel peptide-based tracers anticipate promising future usage not only in diagnostics but also in therapeutic fields.

In PET imaging, scandium-44 (^44^Sc) is a novel radiometal that is of much hope as a potential radioisotope in radiopharmaceutical development and molecular diagnostics. Approximately, in the last 10 years, the number of ^44^Sc-labelled radiopharmaceuticals has been increasing due to their outstanding physicochemical properties such as longer half-life (approximately 4 h), high positron branching (I = 94.27%, E_mean_ (β^+^) = 0.63 MeV), and Lu-like coordination chemistry [18]. A vast array of peptide-based ^44^Sc-labelled radiopharmaceuticals is of pivotal significance in tumor detection (^44^Sc-DOTA-folate, ^44^Sc-DOTA-NOC, ^44^Sc-NODAGA-NOC and ^44^Sc-DOTA-NAPamide), in the monitoring of tumor-associated angiogenetic processes (^44^Sc-AAZTA-RGD), or in hypoxia detection (^44^Sc-labelled DO3AM-NI) [19,20,21,22,23]. Moreover, the favorable biodistribution of ^44^Sc-labelled peptides draws attention to their suitability for the production of GRPR-specific peptide radiopharmaceuticals [24].

Peptide radiolabeling is performed with the application of bifunctional chelators (BFC) [7]. Besides the most widely utilized DTPA or DOTA chelators, studies applying NOTA and NODAGA for labelling purposes are also underway [7]. Based on existing literature data, both NOTA and NODAGA demonstrated superior performance to DOTA during ^68^Ga labeling regarding specific activity, stability, and biodistribution in vivo [25,26]. In one study, the radiochemical characteristics of DOTA, NOTA, and NODAGA were compared when labelling AMBA [7]. Among the evaluated AMBA-chelators, NODAGA-AMBA—labelled with ^68^Ga—was depicted with the most suitable radiochemical properties [7]. Although, in-vitro and in-vivo studies dealing with ^64^Cu and ^18^F-labelled NODAGA-AMBA revealed some of their undesirable characteristics such as relatively low stability and fast tumor clearance [2].

Initiated by the above-detailed research findings, in this study we intended to synthesize AMBA-based NODAGA-conjugated radiocomplexes labeled with both ^68^Ga and ^44^Sc. Furthermore, we aimed at assessing the GRPR specificity of the newly synthetized ^44^Sc-labelled NODAGA-AMBA in vitro and in vivo using PC-3 xenograft prostate tumors.

## 2. Results

### 2.1. Radiochemistry

^68^Ga and ^44^Sc radiolabeling of the NODAGA-AMBA was performed manually behind the L-Block Shield in both cases. The average reaction time of radiolabeling was approximately 25 min (Figure 1). The RCP of both products was found over 98.0%. The molar activity was 19.72 ± 0.13 GBq/µmoL for [^68^Ga]Ga-NODAGA-AMBA and 20.87 ± 0.12 GBq/µmoL for [^44^Sc]Sc-NODAGA-AMBA. 

### 2.2. LogP and Serum Stability Measurements

The octanol/water partition coefficient was found to be −2.75 ± 0.18 for [^68^Ga]Ga-NODAGA-AMBA and −2.81 ± 0.14 for [^44^Sc]Sc-NODAGA-AMBA, showing an insignificant effect of the metal on the polarity of the tracer. For in-vitro stability measurements, the labelled compounds were mixed with mouse plasma, Na_2_EDTA, and oxalic acid. Samples were injected at different time points to the HPLC with and without a column. The comparison of the radioactivity peaks detected during the bypass and the on-column measurements showed no adsorption on the system. In the case of [^68^Ga]Ga-NODAGA-AMBA, the analytical radio-HPLC showed that the RCP of ^68^Ga-labelled compound decreased to approximately 92% at 15 min and decreased to approximately 85% at 90 min. In the case of [^44^Sc]Sc-NODAGA-AMBA, the RCP remained over 98%, which means that the ^44^Sc-labelled compound remained stable during the measured 15 and 90 min time periods.

### 2.3. In Vitro Cellular Uptake Studies

The GRPR specificity of [^68^Ga]Ga-NODAGA-AMBA and [^44^Sc]Sc-NODAGA-AMBA was investigated using receptor-positive PC-3 and negative HaCaT cell lines. The accumulation of [^68^Ga]Ga-NODAGA-AMBA and [^44^Sc]Sc-NODAGA-AMBA in PC-3 cancer cells was significantly higher *(p* < 0.01) than in the receptor negative cell line at each investigated time point (Figure 2). The accumulation of [^68^Ga]Ga-NODAGA-AMBA and [^44^Sc]Sc-NODAGA-AMBA in PC-3 cells was approximately 25-fold higher at 60 and 120 min, than the uptake of the receptor negative HaCaT cells, confirming the GRPR specificity of the investigated radiotracers. Comparing the cellular uptake of the two GRPR specific radiotracers, we found that the [^44^Sc]Sc-NODAGA-AMBA accumulation in PC-3 cells was relatively higher (5.65 ± 0.95 at 60 min; 5.58 ± 1.20 at 120 min) than that of [^68^Ga]Ga-NODAGA-AMBA (4.11 ± 0.79 at 60 min; 3.77 ± 1.08 at 120 min); however, these differences were not significant at *p* < *0.05*. Analyzing the %ID data of the blocking experiments, we found that in the presence of 200 nM BBN during the incubation time, the radiotracer uptake of the GRPR positive PC-3 cells significantly decreased ([^68^Ga]Ga-NODAGA-AMBA: 0.35 ± 0.09 at 60 min and 0.33 ± 0.12 at 120 min; [^44^Sc]Sc-NODAGA-AMBA: 0.38 ± 0.10 and 0.41 ± 0.12 at 60 and 120 min, respectively). In the case of the receptor-negative HaCaT cells, no effect was found after the addition of blocking agent (Figure 2).

### 2.4. Biodistribution and Pharmacokinetic Studies in Healthy Mice

For the determination of the normal distribution of the GRPR-specific radiopharmaceuticals, ex-vivo studies were performed for 30, 60, 120, and 180 min after the intravenous injection of [^68^Ga]Ga-NODAGA-AMBA and [^44^Sc]Sc-NODAGA-AMBA using healthy control animals (Figure 3). After the quantitative analysis of the ex-vivo results, relatively lower [^44^Sc]Sc-NODAGA-AMBA (Figure 3B) accumulation was observed in the selected organs and tissues than using the ^68^Ga-labelled probe (Figure 3A) at the same time points, but this difference was not significant at *p* < 0.05 (Figure 3). Low radiotracer uptake was found in the blood, liver, spleen, gastrointestinal tract, and in the thoracic organs using both radiotracers at each investigated time point. In contrast, notable accumulation was observed in the urinary system (approx. %ID/g: 2–8 in the kidneys, and approx. %ID/g: 250–450 in the urine), in the adrenal glands (approx. %ID/g: 0.5–3) and in the pancreas (approx. %ID/g: 1–4) with both radiotracers. Overall, the radioactivity of the examined organs decreased with time using both ra-diopharmacons (Figure 3). Pharmacokinetics of [68Ga]Ga-NODAGA-AMBA and [44Sc]Sc-NODAGA-AMBA was studied in healthy control CB17 SCID mice. There was no significant difference (at *p*≤0.05) between the pharmacokinetic parameters of [^68^Ga]Ga-NODAGA-AMBA (Figure 3C) and [^44^Sc]Sc-NODAGA-AMBA (Figure 3D). The half-life of the radiolabeled pharmacons in the blood is less than 30 minutes in both cases, and this result is consistent with the log*P* values. The in-vivo stability was determined using healthy mice by analytical radio-HPLC method. Samples were taken at 30, 60, 120, and 180 min and both radiolabeled compounds: [^68^Ga]Ga-NODAGA-AMBA and [^44^Sc]Sc-NODAGA-AMBA remained stable during the measured period. No measurable amount of metabolite was found with radio-HPLC technique, indicating excellent in-vivo metabolic stability.

### 2.5. PET Imaging and Ex Vivo Biodistribution Studies of PC-3 Tumor-Bearing Mice

The GRPR positive tumor-targeting potential of [^68^Ga]Ga-NODAGA-AMBA and [^44^Sc]Sc-NODAGA-AMBA was investigated by preclinical PET imaging 60 and 120 min after intravenous radiotracer injection. Representative decay-corrected PET images are shown in Figure 4. Qualitative analysis of the PET images revealed that the subcutaneously growing GRPR-positive PC-3 tumors were clearly visualized using both radiotracers at each investigated time point (Figure 4A, red arrows). After the quantitative SUV analysis, we found that 60 min after the injection of [^44^Sc]Sc-NODAGA-AMBA, the SUVmean, SUVmax, T/M SUVmean, and T/M SUVmax values of PC-3 tumors were 0.90 ± 0.17, 1.54 ± 0.18, 6.16 ± 1.24, and 6.71 ± 1.08, respectively. Relatively lower accumulation was found in the same tumors by using the ^68^Ga-labelled probe (SUVmean: 0.69 ± 0.15, SUVmax: 1.19 ± 0.11, T/M SUVmean: 5.50 ± 0.54, T/M SUVmax: 5.58 ± 1.07), however, this difference between the two radiotracers was not significant *(p* ≤ 0.05) at 60 min; moreover, it remained the same at 120 min (Figure 4B). Furthermore, 120 min post-injection of [^68^Ga]Ga-NODAGA-AMBA and [^44^Sc]Sc-NODAGA-AMBA the T/M ratios showed higher values than 60 min after injection due to the decreased background activity (Figure 4B, right). This in-vivo data correlated well with the ex-vivo experiments (Table 1), where similarly higher [^44^Sc]Sc-NODAGA-AMBA uptake was observed in PC-3 tumors.

The GRPR receptor specificity of the radiolabeled probes was attested in vivo (Figure 5) and ex vivo (Table 1) by blocking experiments using PC-3 tumor-bearing mice. Assessing the qualitative analysis of the decay-corrected PET images, we found low or moderate accumulation in the experimental PC-3 tumors after 30 min of BBN pretreatment using both radiotracers (Figure 5A, black arrows). The quantitative image analysis showed that the SUV values significantly (*p* ≤ 0.01) decreased (SUVmean: 0.14 ± 0.08, SUVmax: 0.25 ± 0.05 using [^44^Sc]Sc-NODAGA-AMBA; and SUVmean: 0.09 ± 0.05, SUVmax: 0.19 ± 0.07 using [^68^Ga]Ga-NODAGA-AMBA) after intravenous bombesin pretreatment (Figure 5B). When the SUV values of the blocked PC-3 tumors were compared with the non-blocking tumors 120 min after radiotracer injection, we found approximately 6-fold lower accumulation (significant at p ≤ 0.01) using [^68^Ga]Ga-NODAGA-AMBA and [^44^Sc]Sc-NODAGA-AMBA, confirming the GRPR-binding specificity of the radiotracers. These in-vivo data correlated well with the ex-vivo blocking experiments (Table 1). As Table 1 shows, the %ID/g values significantly (*p* ≤ 0.01) decreased in PC-3 tumors using both GRPR-specific radiotracers after the BBN pretreatment, in which observation showed that the tracer uptake of the tumors was blocked efficiently confirming the GRPR-binding specificity (Table 1).

## 3. Discussion

GRPR-overexpressing PCa cells serve as a highly promising area of research in terms of diagnostic advances of PCa [4]. Concerning peptide-based radiopharmaceuticals, considerable attention has been placed on GRPR affine BBN and its analogues thereof with regard to radiopharmaceutical development for imaging and therapeutic purposes as well [27,28]. Recent studies have analyzed the performance of a wide variety of radiolabeled BBN analogues such as AMBA in both diagnostic and therapeutic settings [28].

Therefore, we evaluated the GRPR specificity of two peptide-based radiopharmaceuticals—[^68^Ga]Ga-NODAGA-AMBA and [^44^Sc]Sc-NODAGA-AMBA—applying receptor-positive PC-3 and receptor-negative HaCaT cell lines. Given that both tracer uptake of the PC-3 cells was significantly higher compared to the control at each investigated time point—in accordance with the in-vivo and ex-vivo experiments—we managed to establish the GRPR specificity of the examined radiopharmaceuticals.

Several previous studies have already strengthened the GRPR positivity of PC-3 human prostate cell lines. In recent research conducted by Bologna and co-workers, BBN binding sites were identified in PC-3 cells [29]. In addition, BBN antagonist RC-3095 resulted in the growth inhibition of PC-3 cells transplanted into nude mice [30]. In one study conducted by Liolios et al., PC-3 tumors showed high accumulation of GRPR affine ^99m^Tc-labelled BBN analogue ^99m^Tc-GGC-(Ornithine)3-BN(2-14) (^99m^Tc-BN-O) [31]. Based on the absence of activity of irrelevant molecules including [D-Trp6]LHRH and somatostatin analogue RC-160 to impede the binding of ^125^I[Tyr4]-BBN, BBN receptor specificity of both PC-3 and DU-145 human prostate cell lines was further highlighted by Reile H. and colleagues [6]. In addition, a vast array of preclinical studies proved that PCa models show response to BBN and BBN antagonists [6]. Further, BBN was reported to trigger the proliferation of PC-3 and DU-145 human PCa cell lines [29,32]. Therefore, PC-3 cells seem to be promising to test the diagnostic efficacy of radiolabeled GRPR-peptide analogues.

In our present work we evaluated the GRPR specificity of two peptide-based radiopharmaceuticals: [^68^Ga]Ga-NODAGA-AMBA and [^44^Sc]Sc-NODAGA-AMBA. For this purpose, already-proven GRPR receptor positive PC-3 cells and receptor-negative control HaCaT cell lines were applied. Given that both tracer uptake of the PC-3 cells was significantly higher compared to the control at each investigated time point, in accordance with the in vivo and ex vivo experiments, we managed to establish the GRPR specificity of the examined radiopharmaceuticals.

In agreement with our results, according to a prior study executed by Zhang-Yin and co-workers, PC-3 tumors were visualized with increased ^68^Ga-AMBA uptake [33]. With the application of PC-3 PCa cell lines characterized by GRPR expression but androgen receptor (AR) and prostate specific membrane antigen (PSMA) deficiency, Schroeder et al. reported the superiority of ^68^Ga-AMBA over ^18^F-fluorocholine (^18^F-FCH) regarding the in-vivo PET imaging of PCa xenografts [16]. Further, PC-3 tumors were definitely recognized with [^68^Ga]Ga-NOTA-AMBA in μPET/CT images of PC-3 xenograft mice in research carried out by Dam et al. [34]. The above-detailed results together with no lacrimal or salivary gland accumulation and insignificant hepatobiliary elimination of ^68^Ga-AMBA presuppose the feasibility of ^68^Ga-labelled peptide-based PET imaging of PCa with enhanced contrast and a more tissue-specific targeting potential. Although our results of ^68^Ga-GRPR-based peptide imaging are comparable to those of the existing literature, to our best knowledge, no previous research data is available regarding the performance evaluation of [^44^Sc]Sc-NODAGA-AMBA in the diagnostics of PCa.

In vitro cellular uptake analysis of the two GRPR-specific radioconjugates revealed that the accumulation of [^44^Sc]Sc-NODAGA-AMBA in PC-3 cells was relatively higher than that of [^68^Ga]Ga-NODAGA-AMBA, although the difference did not seem to be statistically significant. The explanation of this finding is not yet fully covered, and future large scale studies are required to elucidate the exact reason behind this. Since the 4-hour physical half-life of ^44^Sc makes the transportation of ^44^Sc-labelled tracers to distinct isotope laboratories possible, physiological processes featured with slower kinetic properties could be widely assessed [35,36]. Additionally, improved resolution and image quality of ^44^Sc PET images compared to ^68^Ga PET scans may further outline the excellence of ^44^Sc over ^68^Ga [35]. Even though, head-to-head comparison of the uptake of ^68^Ga and ^44^Sc-labeled peptides did not reveal any remarkable differences, the favorable characteristics of ^44^Sc for radiolabeling procedures may emphasize its superiority over ^68^Ga in the diagnostic algorithm of PCa.

Applying GRPR-blocking experiment with BBN, we managed to confirm the specific tumor-targeting efficiency of the investigated radioconjugates. In accordance with in-vivo and ex-vivo studies, significantly decreased tracer uptake of the GRPR-positive PC-3 cells was depicted in the case of the blocking experiments, whereas the blocking agent expressed no effect on control HaCaT cells. In line with our results, Kim et al. reported discrete tumor uptake of [^64^Cu]Cu-NODAGA-BBN or [^64^Cu]Cu-NODAGA-galacto-BBN following the administration of 15 mg/BW of non-radioactive blocking ligands (NODAGA-BBN, or NODAGA-galacto-BBN) [37]. They presented the reduction of the radiopharmaceutical accumulation in the group of PC-3 tumor-bearing nude mice injected with the GRPR-blocking agent 30 min prior to the tracer administration compared to those pets that did not receive the blocking ligand [37].

Prior research generally confirms that GRPR could be encountered amongst others in the central nervous system, gastrointestinal (GIT) tract, pancreas, and the adrenal cortex [5]. Therefore, we employed healthy control animals to explore the physiological biodistribution of GRPRs. In accordance with literature data substantiating the physiological existence of GRPR in urogenital smooth muscle, we depicted considerable tracer uptake in the urinary system as well [38]. Radiopharmaceutical accumulation in the kidneys and in the urine remained high due to the predominant renal elimination of the tracer. In line with our ex-vivo results, Fournier et al.—evaluating two BBN-based radiopharmaceuticals for the identification of breast and prostate cancers in *Balb/c* and tumor-bearing *Balb/c* nude mice—showed elevated accumulation of both ^64^Cu and ^68^Ga/NOTA-PEG-[D-Tyr^6^,βAla^11^,Thi^13^,Nle^14^]BBN(6-14) [^68^Ga/NOTA-PEG-BBN(6-14)] in the pancreas and adrenal glands abounding in GRPR [39,40]. Another study also corroborated the presence of BB2 (bombesin type 2 receptor/GRPR) in pancreatic acinar cells [41]. Further, Liolios and co-workers showed notably increased uptake of GRPR affine [^99m^Tc]Tc-BN-O of the pancreas compared to other tissues [31]. However, in agreement with our findings, low (below 6% ID/g) tracer accumulation was depicted in the heart, lung, muscle, and spleen [31]. Further supporting the occurrence of GRPR in the lung, Johnson et al. reported that pulmonary neuroendocrine cells (NECs) exhibited the gene of GRPR [42]. Moreover, a high amount of GRP was depicted in both human and mouse fetal lung [43,44]. In addition, healthy control animals in our study demonstrated low gastrointestinal [^68^Ga]Ga-NODAGA-AMBA and [^44^Sc]Sc-NODAGA-AMBA uptake. Existing literature data also evidenced minor GRPR expression in the neuroendocrine cells of the gastrointestinal organs [45].

In summary, both [^68^Ga]Ga-NODAGA-AMBA and [^44^Sc]Sc-NODAGA-AMBA express outstanding tumor-targeting PET properties. Given the advantageous chemical characteristics of ^44^Sc, [^44^Sc]Sc-NODAGA-AMBA seems to be a novel clinically translatable BBN analogue-based PET radiopharmaceutical in the diagnostic assessment of GRPR positive PCa.

## 4. Materials and Methods

### 4.1. Chemicals

NODAGA-Gly-4-Abz-Gln-Trp-Ala-Val-Gly-His-Leu-Met-NH_2_ (NODAGA-AMBA) was obtained from ABX advanced biochemical compounds GmbH (Cat. No.: 9814) (Radeberg, Germany). For the radiolabeling procedures, the ultra-pure (u.p.) solvents and sodium acetate (NaOAc) were obtained from Sigma-Aldrich Ltd. (Budapest, Hungary). Ultra-pure HCl was purchased from Merck Ltd. (Budapest, Hungary). ^68^Ga radioisotope was obtained from a ^68^Ge/^68^Ga isotope generator (Gallia-Pharm, Eckert and Ziegler Germany), ^44^Sc was produced in a GE PETtrace cyclotron at the Division of Nuclear Medicine and Translational Imaging, Department of Medical Imaging, Faculty of Medicine, University of Debrecen (Debrecen Hungary). All other reagents and solvents were obtained from Sigma-Aldrich Ltd. (Budapest, Hungary) and VWR International Ltd. (Debrecen, Hungary) and used without further purification.

### 4.2. ^68^Ga-Labelling of NODAGA-AMBA

^68^Ga (t_½_ = 68 min, β^+^ = 89% and EC = 11%) was acquired from a ^68^Ge/^68^Ga generator (50 mCi, Gallia-Pharm, Eckert and Ziegler, Berlin, Germany). The labelling protocol is based on our previous work [46]. Briefly, the generator eluate was fractioned and the top fraction that contained 70–75% of the total radioactivity was used for radiolabeling. An amount of 1,2 mL of ^68^Ga-solution from the highest activity aliquot, 170 μL NaOAc buffer (0.5 M, pH = 4), 70 μL NaOH (2%), and 6 μL of stock solution NODAGA-AMBA (1 mM) were mixed to ensure a pH of 4.3−4.5 in a 5 mL Eppendorf tube, and this was incubated for 5 min at 95 °C. Thereafter, the solution was pipetted into an Oasis HLB 1 cc Vac Cartridge (Waters) and was washed with 2 mL of water to remove a buffer. The product ([^68^Ga]Ga-NODAGA-AMBA) was eluted with 350 μL of 96% EtOH/isotonic NaCl solution (mixture ratio 2:1). The radiochemical purity (RCP) of the product was determined with radio-HPLC on a KNAUER RP-HPLC system with the Supelco Discovery^®^ Bio Wide Pore C-18 analytical column (250 mm × 4.6 mm; particle size: 10 μm). The HPLC system was combined with a radiodetector and the signals were detected simultaneously. Gradient elution was achieved at a flow rate of 1 mL/min. The mobile phase consisted of eluent A: (0.1% TFA in water) and eluent B: (0.1% TFA in acetonitrile-water (95:5, *v*/*v*)). Before performing further experiments, the product was diluted with isotonic (0.9%) saline and was filtered to be sterile.

### 4.3. ^44^Sc-Labelling of NODAGA-AMBA

One-hundred-and-twenty milligrams of natural calcium (Ca) (99.99%) as a solid target was pressed into an aluminum target holder and was irradiated for 60 min with 30 μA beam using the GE PETtrace Cyclotron. After the irradiation, the irradiated Ca target was dissolved in 3 M u.p. HCl and the solution was transferred into a preconditioned self-loaded DGA resin (70 mg/cartridge). Following the loading of solution, the column was washed with 3 mL 3 M u.p. HCl, and 3 mL 1 M HNO_3_, and then it was repeatedly washed with 3 mL 3 M u.p. HCl to remove the remaining Ca target materials. After the purification, the ^44^Sc isotope was eluted with 2 mL 0.1 M u.p. HCl in 200 μL fractions, and the highest activity fractions were mixed and were used for radiolabeling. The labelling protocol is based on our previous work [21]. Briefly: 1 mL ^44^Sc-solution, 1150 μL NaOAc buffer (0.5 M, pH = 4), and 6 μL of stock solution of NODAGA-AMBA (1 mM) were mixed, and the reaction was incubated for 15 min at 95 °C. Thereafter, the solution was pipetted into a Light C18 Sep-Pak Cartridge and was washed with 2 mL of water. The product ([^44^Sc]Sc-NODAGA-AMBA) was eluted with 500 μL of 96% EtOH/isotonic NaCl solution (mixture ratio 2:1). The RCP of the product was determined with the above-mentioned KNAUER RP-HPLC system.

### 4.4. Determination of Partition Coefficient and Metabolic Stability of ^68^Ga- and ^44^Sc-Labelled NODAGA-AMBA

For the determination of Log*P* value and the in-vitro serum stability, the same protocol was used as was described earlier by our research group [47]. Briefly, for the determination of the partition coefficient (log*P*), 10 µL of [^68^Ga]Ga-NODAGA-AMBA or [^44^Sc]Sc-NODAGA-AMBA solution (approximately 5 MBq) was mixed with 500 µL of 1-octanol and 490 µL of water in a test tube. To reach equilibrium state, the mixture was firmly stirred and then centrifuged (20.000 rpm, 5 min). One-hundred microliters of the samples were pipetted into vials from each layer, and the radioactivity of the fractions was determined with a calibrated gamma counter (Perkin-Elmer Packard Cobra, Waltham, MA, USA). The measurements were performed in triplicates for both labelled compounds.

After 15 and 90 min incubation time, a 50 µL sample was taken and 50 µL abs. EtOH was added to the aliquots. Then, the samples were centrifuged (20.000 rpm, 5 min), and the supernatant was removed and diluted with the eluent of HPLC. This was followed by the performance of the analytical measurements.

### 4.5. Cell Lines

Human PCa PC-3 (positive—high BBN/GRPR expression) and human immortal keratinocyte HaCaT (negative) cell lines were purchased from the American Type Culture Collection (ATCC, Manassas, VA, USA) and Thermo Fisher Scientific (London, UK), respectively. PC-3 and HaCaT cells were cultured in RPMI-1640 medium (Sigma-Aldrich, St. Louis, MO, USA) with 10% Fetal Bovine Serum (FBS, GIBCO Life technologies) supplemented with 1% Antibiotic and Antimicotic solution (Sigma-Aldrich). All cell lines were cultured at standard conditions (5% CO_2_, 37 °C). For in-vitro uptake, measurements and subcutaneous tumor inoculation cells were used at 85% confluence after five passages. The viability of the cells was always higher than 90%, as assessed by the trypan blue exclusion test.

### 4.6. Cellular Uptake Studies

PC-3 and HaCaT cells were trypsinized, centrifuged, suspended, and aliquoted in test tubes at a cell concentration of 1 × 10^6^/1 mL RPMI-1640 solution. Tubes were incubated for 60 or 120 min in the presence of 0.37 MBq of [^68^Ga]Ga-NODAGA-AMBA or [^44^Sc]Sc-NODAGA-AMBA at 37 °C. In blocking experiments, 200 nM bombesin (Sigma-Aldrich) was added to the cells. After the incubation time, samples were washed three times with ice-cold PBS and the radioactivity was measured with a calibrated gamma counter (Perkin-Elmer Packard Cobra, Waltham, MA, USA) for 1 min within the ^68^Ga- and ^44^Sc-sensitive energy window. Decay-corrected radiotracer uptake was expressed as counts/(min × (10^6^ cells)) (cpm). The uptake of the radiopharmaceuticals was expressed as percentage of the total radioactivity of radiotracers added to the cells (%ID/million cells). Each experiment was performed in triplicate and the displayed data represent the means of at least three independent experiments (±SD).

### 4.7. In Vivo Tumor Model

Immunodefficient CB17 SCID mice were housed under sterile conditions in IVC cages (Sealsafe Blue line IVC system, Techniplast, Akronom Ltd., Budapest, Hungary) at the temperature of 26 ± 2 °C with 55 ± 10% humidity and artificial lighting with a circadian cycle of 12 h. Sterile semi-synthetic diet (Akronom Ltd., Budapest, Hungary) and sterile drinking water were available ad libitum to all animals. Laboratory animals were kept and treated in compliance with all applicable sections of the Hungarian Laws and regulations of the European Union.

For animal experiments, 12-week-old female CB17 SCID (n = 64) were used. For the induction of GRPR-expressing tumor model, mice were anesthetized with a dedicated small animal anesthesia device (Tec3 Isoflurane Vaporizer, Eickemeyer Veterinary Equipment, Sunbury-on-Thames, UK) applying 3% isoflurane (Forane, AbbVie), 0.4 L/min O_2_, and 1.4 L/min N_2_O, and 5 × 10^6^ PC-3 tumor cells in 0.9% NaCl (100 µL) were injected subcutaneously into the left shoulder area of CB17 SCID mice. In-vivo and ex-vivo experiments were carried out 14 ± 1 days after subcutaneous injection of tumor cells at the tumor volume of approximately 86 mm^3^.

### 4.8. In Vivo PET Imaging

For in-vivo imaging studies, mice were injected intravenously with 11.3 ± 1.4 MBq of [^68^Ga]Ga-NODAGA-AMBA or [^44^Sc]Sc-NODAGA-AMBA via the lateral tail vein under isoflurane anesthesia. Sixty and 120 min after radiotracer injection, mice were anesthetized by 3% isoflurane (Forane) and 20 min static PET scans were performed from the tumorous area using the preclinical miniPET device (University of Debrecen, Faculty of Medicine, Department of Medical Imaging, Division of Nuclear Medicine and Translational Imaging). Following the reconstruction of PET volumes using the three-dimensional Ordered Subsets Expectation Maximization (3D-OSEM) algorithm, volumes of interest (VOIs) were manually drawn around the examined regions using the BrainCAD image analysis software and quantitative standardized uptake values (SUVs) values were calculated as follows: SUV = [VOI activity (Bq/mL)]/[injected activity (Bq)/animal weight (g)], assuming a density of 1 g/mL. Tumor-to-muscle (T/M) ratios were calculated from the SUV values of the tumor and background (muscle).

### 4.9. Ex Vivo Biodistribution Studies

Thirty, 60, 120, and 180 min after the intravenous injection of 11.3 ± 1.4 MBq [^68^Ga]Ga-NODAGA-AMBA or [^44^Sc]Sc-NODAGA-AMBA healthy control and PC-3 tumor-bearing mice were euthanized with 5% Forane, sacrificed, and tissue samples were taken from the selected organs. The weight and the radioactivities of both the tumors and normal tissues were measured with calibrated gamma counter, and the uptake was expressed as %ID/g tissue.

### 4.10. Pharmacokinetic Studies and In Vivo Stability

For pharmacokinetic studies, healthy control CB17 SCID mice (n = 12) were injected intravenously with 10.8 ± 1.6 MBq of [^68^Ga]Ga-NODAGA-AMBA or [^44^Sc]Sc-NODAGA-AMBA under isoflurane anesthesia. Thereafter, approximately 30 μL blood samples were collected from the saphenous vein into a capillary tube at the following time points: 30, 60, 120, and 180 minutes. The volume of the blood was determined using a digital caliper. Blood samples were placed in a γ-counter and the radioactivity of each sample was measured. Results were expressed as a percentage of the injected activity per mL (% ID/mL). For the determination of in vivo serum stability of [^68^Ga]Ga-NODAGA-AMBA and [^44^Sc]Sc-NODAGA-AMBA, blood samples were taken from the mice at the previously mentioned time points (30, 60, 120, and 180 min). Firstly, the blood samples were centrifuged at 4°C, 10,000 rpm for 5 min. Thereafter, 50 μL samples were taken from the supernatant and mixed with ice-cold abs. ethanol (50 μL) and centrifuged again at 4°C, 10 000 rpm for 5 min. The supernatants were analyzed by analytical radio-HPLC. In all cases, the radio-HPLC chromatograms were compared to the original chromatograms of the radiotracers to find any metabolite forms.

### 4.11. Blocking Experiments

For blocking experiments, PC-3 tumor-bearing mice were injected intravenously with 15 mg/kg of BBN (Sigma-Aldrich) 30 min prior to the injection of 11.3 ± 1.4 MBq [^68^Ga]Ga-NODAGA-AMBA or [^44^Sc]Sc-NODAGA-AMBA and in-vivo and ex-vivo organ distribution studies were performed as described above.

### 4.12. Statistical Analysis

Significance was calculated by student’s two-tailed *t*-test, two-way ANOVA, and the Mann–Whitney rank-sum tests, and the significance level was set at *p* ≤ 0.05 unless otherwise indicated. A commercial software package (MedCalc 18.5, MedCalc Software, Mariakerke, Belgium) was used for all statistical analyses. Data are presented as mean ± SD of at least three independent experiments.

## 5. Conclusions

In conclusion, our newly synthesized [^44^Sc]Sc-NODAGA-AMBA radiopharmaceutical showed excellent binding affinity to GRPR-positive PC-3 prostate cancer cells and tumors. Due to its favorable physical-chemical properties and high selectivity, [^44^Sc]Sc-NODAGA-AMBA seems to be a promising molecular probe for PET imaging of PSMA and AR-negative prostate cancers and metastases.

## Figures and Tables

**Figure 1 ijms-23-10061-f001:**
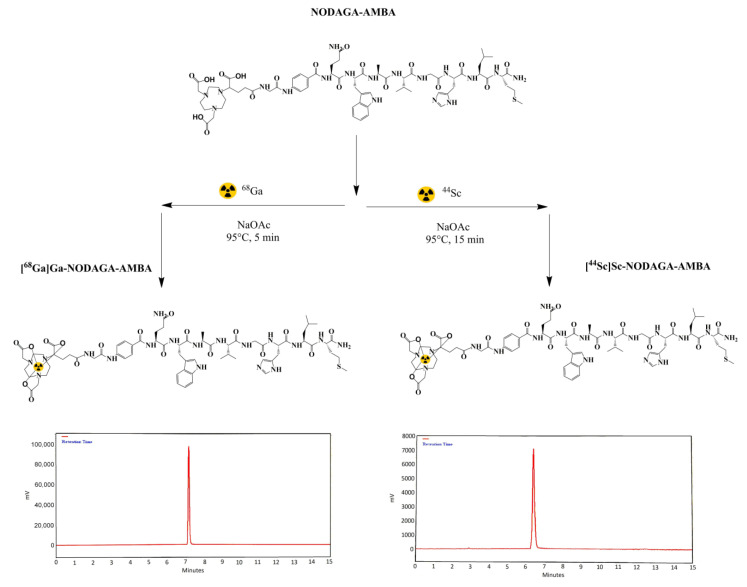
Schematic representation of ^68^Ga and ^44^Sc radiolabeling reactions of the NODAGA-Gly-4-Abz-Gln-Trp-Ala-Val-Gly-His-Leu-Met-NH_2_ (NODAGA-AMBA) precursor and radio-HPLC chromatogram of the [^68^Ga]Ga-NODAGA-AMBA and [^44^Sc]Sc-NODAGA-AMBA.

**Figure 2 ijms-23-10061-f002:**
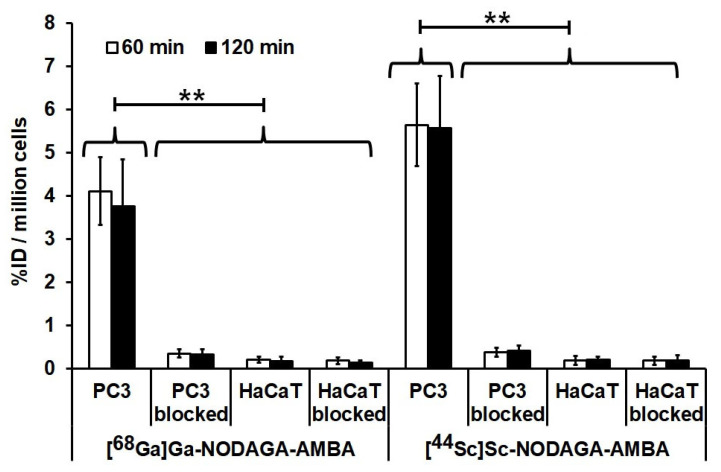
Assessment of in-vitro cellular uptake studies of gastrin-releasing peptide receptor (GRPR)-specific [^68^Ga]Ga-NODAGA-AMBA and [^44^Sc]Sc-NODAGA-AMBA radiopharmaceuticals. Comparison of radiotracer uptake results of GRPR-positive PC-3 and negative HaCaT cells after 60 and 120 min incubation time in the presence and absence of 200 nM bombesin (BBN) as a blockade. Significance level between the PC-3 cells and the blocked PC-3 or HaCaT cells: *p* ≤ 0.01 (**). %ID: Radiotracer accumulation in 10^6^ cells was expressed as the percentage of the incubating dose. The data shown are means ± SD of the results of at least three independent experiments, each performed in triplicate.

**Figure 3 ijms-23-10061-f003:**
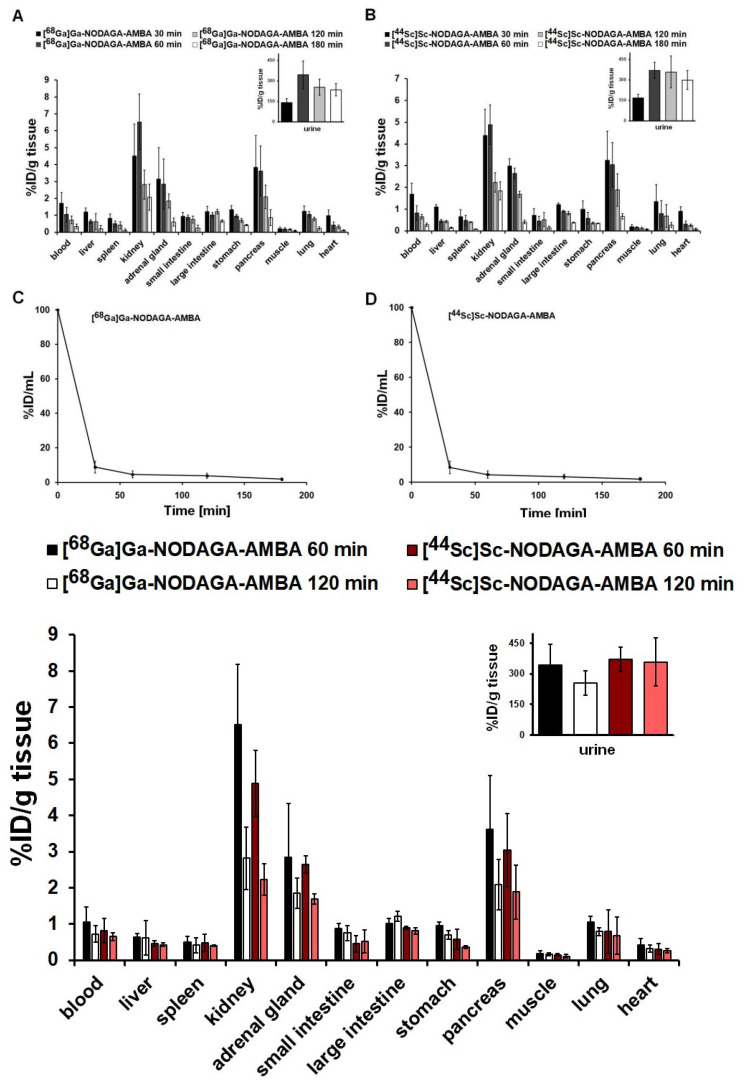
Ex-vivo biodistribution data for [^68^Ga]Ga-NODAGA-AMBA (**A**) and [^44^Sc]Sc-NODAGA-AMBA (**B**). Quantitative %ID/g tissue analysis of ex-vivo biodistribution data (n = 5 control animals/radiotracer/time point) 30, 60, 120, and 180 min after intravenous tracer injection. %ID values are presented as mean ± SD. In-vivo blood clearance of [^68^Ga]Ga-NODAGA-AMBA (**C**) and [^44^Sc]Sc-NODAGA-AMBA (**D**) in healthy control CB17 SCID mice (n = 3 animals/radiotracer/time point). %ID/mL values are presented as mean ± SD.

**Figure 4 ijms-23-10061-f004:**
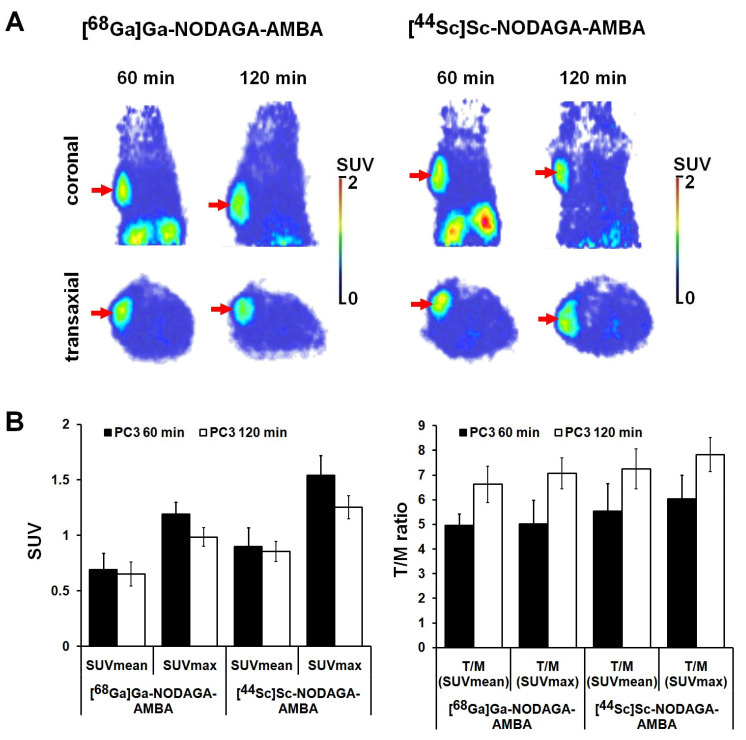
In-vivo assessment of tumor-targeting properties of [^68^Ga]Ga-NODAGA-AMBA and [^44^Sc]Sc-NODAGA-AMBA radiotracers. (**A**) positron emission tomography (PET) imaging and quantitative image analysis of PC-3 tumors [^68^Ga]Ga-NODAGA-AMBA and [^44^Sc]Sc-NODAGA-AMBA radiotracers. (**A**) Representative coronal (**upper row**) and transaxial (**lower row**) decay-corrected PET images of GRPR-positive PC-3 tumor-bearing mice 60 and 120 min post-injection, and 14 ± 1 days after tumor cell inoculation. (**B**) quantitative standardized uptake value (SUV) analysis of [^68^Ga]Ga-NODAGA-AMBA and [^44^Sc]Sc-NODAGA-AMBA accumulation in experimental PC3 tumors (n = 5 animals/radiotracer/time point). Red arrows: PC3 tumors; T/M: tumor-to-muscle ratio. SUV values are presented as mean ± SD.

**Figure 5 ijms-23-10061-f005:**
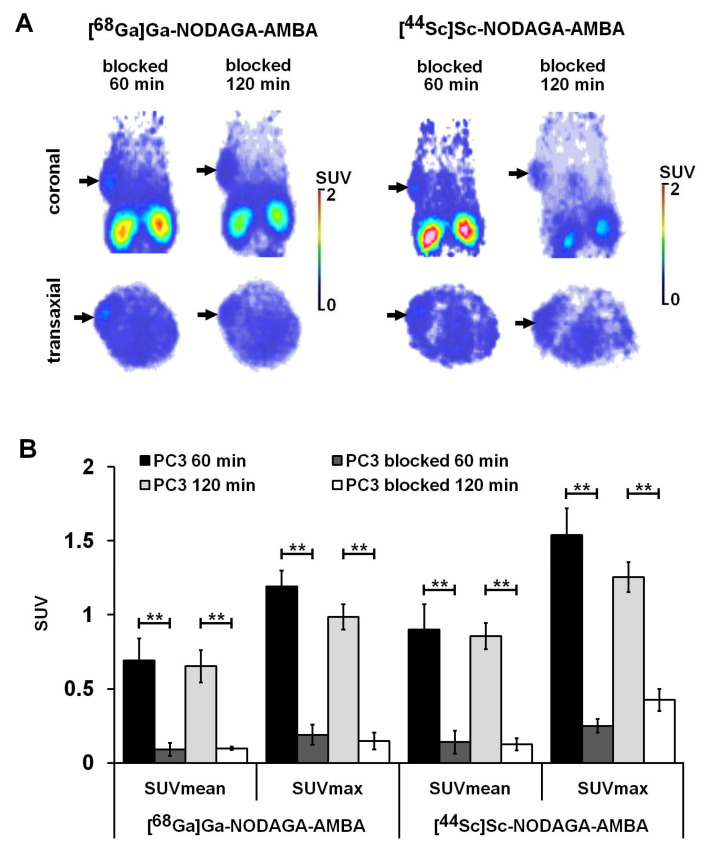
In-vivo PET imaging and quantitative image analysis of subcutaneous PC-3 using [^68^Ga]Ga-NODAGA-AMBA and [^44^Sc]Sc-NODAGA-AMBA radiotracers. (**A**) Representative coronal (upper row) and transaxial (lower row) PET images of blocked (15 mg/kg bombesin) GRPR-positive PC-3 tumors. (**B**) Quantitative SUV analysis of [^68^Ga]Ga-NODAGA-AMBA and [^44^Sc]Sc-NODAGA-AMBA accumulation in blocked PC-3 tumors (n = 5 animals/radiotracer/time point). Decay-corrected PET images and data were obtained 14 ± 1 days after tumor cell inoculation and 60 and 120 min after intravenous injection of the radiotracers. Black arrows: blocked PC3 tumors. Significance level: *p* ≤ 0.01 (**). T/M: tumor-to-muscle ratio. SUV values are presented as mean ± SD.

**Table 1 ijms-23-10061-t001:** Ex-vivo assessment of [^68^Ga]Ga-NODAGA-AMBA and [^44^Sc]Sc-NODAGA-AMBA accumulation (%ID/g) in PC-3 experimental tumors 60 and 120 min after intravenous tracer injection and 14 ± 1 days after tumor induction. Significance level between non-blocked and blocked tumors: *p* ≤ 0.01 (**); 15 mg/kg BBN was used for blocking. T/M: tumor-to-muscle ratio.

Tumor	[^68^Ga]Ga-NODAGA-AMBA		[^44^Sc]Sc-NODAGA-AMBA	
	60 min	120 min	60 min	120 min
PC3	3.78 ± 0.93 **	3.29 ± 1.20 **	4.56 ± 0.45 **	4.14 ± 0.47 **
PC3 blocked	0.60 ± 0.22	0.48 ± 0.14	0.79 ± 0.16	0.69 ± 0.17
PC3 T/M	13.21 ± 2.47 **	21.64 ± 3.78 **	16.88 ± 1.96 **	27.57 ± 2.88 **
PC3 T/M blocked	1.97 ± 0.22	2.86 ± 0.47	2.19 ± 0.36	3.04 ± 0.62

## Data Availability

The dataset used and/or analyzed during the current study are available from the corresponding author on reasonable request.

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
