# Peer review of "PET Probes for Preclinical Imaging of GRPR-Positive Prostate Cancer: Comparative Preclinical Study of [68Ga]Ga-NODAGA-AMBA and [44Sc]Sc-NODAGA-AMBA"

_ijms, 2022, doi:10.3390/ijms231710061_

Round 1
Reviewer 1 Report
The manuscript entitled “PET Probes for Preclinical Imaging of GRPR-Positive Prostate 2 Cancer: Comparative Preclinical Study of [68Ga]Ga-NODAGA-3 AMBA and [44Sc]Sc-NODAGA-AMBA” describes the preparation of the new gastrin-releasing peptide receptor (GRPR) overexpressing PC-3 prostate cancer targeted, scandium-44 isotope labeled radiopharmaceutical candidate [44Sc]Sc-NODAGA-AMBA. Authors also provide the new agent’s comparison to its gallium-68-labeled variant ([68Ga]Ga-NODAGA-AMBA) along in vitro and in vivo (PET) studies of prostate tumor (PCa) PC-3 xenografts. The paper is definitely original work that will be of interest to prospective research studies, pharmaceutical developments related to the promising scandium-44 isotope, or to further radiolabeled bombesin analogues and relevant theranostic applications.
Considering the above mentioned valuable and citable elements of the current work, and considering the appropriate research design, the adequately described methods, and the clearly presented results, the paper can be accepted for publish without further corrections.
Author Response
Dear Reviewer 1,
We would like to express our gratitude for your review.
Yours sincerely,
György Trencsényi
corresponding author

Reviewer 2 Report
The article presents the synthesis and in vivo validation of two new GRPR selective tracers. Although the results presented are interesting, the article needs to revise both the text and complementary experiments to support the conclusions.
1.- Although the manuscript presents promising results in the development of new diagnostic agents in the field of cancer, the text needs to improve the writing, especially the abstract and the introduction, which is written more as a discussion than as the state of the art of a topic. In addition, in the introduction section, the authors should provide more information on the new radioisotope used (44Sc), since it is the basis for the work presented. Other radiotracers developed based on this radionuclide, advantages, disadvantages, production…
2.-The bibliography is quite outdated, only two articles are after 2020.
3.- Since the work presents for the first time the synthesis of a new tracer, further characterization of the tracer is necessary. The author should complete its characterization with other techniques such as MS to validate the nature of the peptide or the specific activity.
4.-In vitro stability studies are scarce. It is recommended to perform a longitudinal study including different times (not only one point) to analyze the evolution not only of the isotope release but also the degradation of the peptide. This requires not only the percentage, but also the visualization of the chromatogram (HPLC or ITLC) to observe possible secondary metabolites generated from the degradation.
5.- A more in-depth in vivo study of the pharmacokinetics of the compound is required, evaluating blood circulation time, biodistribution at different times, stability in serum at different times.
6.- A study at the biomolecular level is lacking, both with in vitro studies of tracer uptake and tumor tissue where the presence of the receptor is demonstrated.
7.- Have the synthesis conditions been selected based on previous work? it is advisable to include a synthesis optimization study to confirm that these are the best conditions for the design.
Author Response
Dear reviewer 2,
Please see the attachment.
